# Genomic Characterization and Expression Analysis of the SnRK Family Genes in *Dendrobium officinale* Kimura et Migo (Orchidaceae)

**DOI:** 10.3390/plants10030479

**Published:** 2021-03-03

**Authors:** Yue Wang, Aizhong Liu

**Affiliations:** 1Key Laboratory of Economic Plants and Biotechnology, Yunnan Key Laboratory for Wild Plant Resources, Kunming Institute of Botany, Chinese Academy of Sciences, Kunming 650201, China; wangyue@mail.kib.ac.cn; 2Bio-Innovation Center of DR PLANT, Kunming Institute of Botany, Chinese Academy of Sciences, Kunming 650201, China; 3Key Laboratory for Forest Resources Conservation and Utilization in the Southwest Mountains of China, Ministry of Education, Southwest Forestry University, Kunming 650224, China

**Keywords:** sucrose non-fermenting1-related protein kinases (SnRKs), *Dendrobium officinale*, expression patterns, stress

## Abstract

Sucrose non-fermenting1-related protein kinases (SnRKs) are a type of Ser/Thr protein kinases, and they play an important role in plant life, especially in metabolism and responses to environmental stresses. However, there is limited information on *SnRK* genes in *Dendrobium officinale*. In the present research, a total of 36 *DoSnRK* genes were identified based on genomic data. These DoSnRKs could be grouped into three subfamilies, including 1 member of DoSnRK1, 7 of DoSnRK2, and 28 of DoSnRK3. The gene structure analysis of *DoSnRK* genes showed that 17 members had no introns, while 16 members contained six or more introns. The conserved domains and motifs were found in the same subfamily. The various cis-elements present in the promoter regions showed that *DoSnRK* genes could respond to stresses and hormones. Furthermore, the expression patterns of *DoSnRK* genes in eight tissues were investigated according to RNA sequencing data, indicating that multiple *DoSnRK* genes were ubiquitously expressed in these tissues. The transcript levels of *DoSnRK* genes after drought, MeJA, and ABA treatments were analyzed by quantitative real-time PCR and showed that most *DoSnRK* genes could respond to these stresses. Therefore, genomic characterization and expression analyses provide valuable information on *DoSnRK* genes for further understanding the functions of SnRKs in plants.

## 1. Introduction

Protein kinases are important regulators in response to stresses that can activate downstream genes in different signal pathways through phosphorylation [1,2]. One of the well-known kinases is the sucrose nonfermenting1 (SNF1) kinase that is widely found in living organisms, including the SNF1 protein in yeast, the AMP-activated protein kinase (AMPK) in mammals, and the sucrose non-fermenting1-related protein kinase (SnRK) in plants [2,3]. Studies have showed that SnRKs play vital roles in plant growth and development, especially in stress responses and global metabolism [1,3,4,5,6,7,8,9].

In plants, SnRK proteins can be divided into three subfamilies based on the sequences and structures, including SnRK1, SnRK2, and SnRK3 [10,11]. The SnRK1 subfamily contains a highly conserved protein kinase (Pkinase) domain at the N-terminus, which is the homolog of SNF1 in yeast and AMPKs in mammals [3]. SnRK2 and SnRK3 subfamilies are specific to plants and contain more members than the SnRK1 subfamily. SnRK2 and SnRK3 subfamilies are thought to evolve from SnRK1 and form more functions during evolution [12]. The SnRK2 subfamily contains a kinase domain and an ATP-binding domain. According to the dependence on abscisic acid (ABA), the SnRK2 subfamily can be classified into three groups [2]. The members of group 1 are independent of ABA, the members of group 2 are independent or weakly dependent on ABA, while the members of group 3 are strongly dependent on ABA [2,13]. The SnRK3 subfamily can bind to calcineurin B-like (CBL) proteins and participate in the stress response through the Ca^2+^ pathway, and therefore, SnRK3 proteins are called calcineurin B-like calcium sensor-interacting protein kinases (CIPKs) [2]. Except for a protein kinase domain at the N-terminus, SnRK3 proteins contain Asn-Ala-Phe (NAF) domains and protein–protein interaction (PPI) domains at the C-terminus. The NAF domain is named due to the prominent presence of conserved Asn-Ala-Phe amino acids and is responsible for interactions with CBL proteins, while the PPI domain is responsible for interactions with phosphatase 2C [14].

The functions of SnRK1, SnRK2, and SnRK3 in plants have been investigated and shown to diverge often. SnRK1 is thought to be an essential energy sensor in plants to regulate global metabolism [7,15]. In potato (*Solanum tuberosum* L.), SnRK1 was demonstrated to regulate the gene expression level and activity of sucrose synthase, which showed its important role in carbohydrate metabolism [15]. The decreased activity of SnRK1 in pea (*Pisum sativum*) resulted in more sucrose accumulation, with defects in seed maturation [14]. SnRK1s can directly phosphorylate some enzymes in different metabolic pathways, such as hydroxy-3-methyl glutaryl coenzyme A reductase, sucrose phosphate synthase, and trehalose phosphate synthase [2]. Other functions of SnRK1 have been recently reported [16]. OsSnRK1a conferred broad-spectrum resistance through regulation of the salicylic acid and jasmonic acid pathways in rice [17]. Overexpression of soybean *GmSnRK1.1* resulted in increased resistance to *Phytophthora sojae*, while silencing *GmSnRK1.1* reduced its resistance [18]. In *Arabidopsis thaliana*, SnRK1 positively regulated somatic embryogenesis, seed yield, and growth under high temperature through phosphorylation of FUSCA3, a transcription factor [19]. 

SnRK2 has been extensively studied, and it has been demonstrated that it is involved in the ABA signaling pathway, glucose metabolism, and various stresses [4,6,11,20,21]. In particular, as an essential factor in the ABA pathway, it plays an important role in the regulation of stomatal closure, flowering time, seed development, and dormancy [22,23]. The first reported *SnRK2* gene (PKABA1) was identified in wheat, which was triggered by ABA and dehydration [2]. It has been confirmed that three SnRK2 members (AtSnRK2.2, AtSnRK2.3, and AtSnRK2.6) in *Arabidopsis* participate in the ABA signaling pathway and function in response to water stress [24]. The seedlings of the triple-knockout mutant *snrk2.2/2.3/2.6* were not sensitive to ABA, featuring phenotypes of early flowering and producing less seeds [25]. AtSnRK2.6 (AtOST1) phosphorylates INDUCER OF CBF EXPRESSION1 (ICE1) and enhances its stability and transcription, which results in stronger cold tolerance [9]. Overexpression of *AtSnRK2.6* enhances the metabolism of sucrose and fatty acids in leaves, which facilitates physiological processes [4]. *SnRK2* genes can be induced by ABA, salt, and other stresses [4]. Overexpression of *SnRK2* genes from different species in *Arabidopsis* and tobaccos can improve tolerance to drought and/or salt in transgenic plants [26,27,28,29,30,31,32]. For example, transgenic tobaccos with overexpression of *TaSnRK2.9* showed a significant increase in tolerance to salt and drought stress through stronger abilities to conduct reactive oxygen species (ROS) scavenging, ABA signal transduction, and specific interaction between SnRKs and ABA-responsive element (ABRE)-binding factor (ABF) [33].

SnRK3 proteins can interact with CBL proteins to form a calcium signaling complex, which plays an important role in response to various stresses, including salt, drought, cold, and ABA [34]. The most famous function of SnRK3s (CIPKs) is as important components in the salt overly sensitive (SOS) pathway in *Arabidopsis.* Salt overly sensitive2 (SOS2; AtSnRK3.11) and SOS3 (CBL4) interact to form a complex, which can moderate the gene expression and activity of SOS1, a Na+/H+ exchanger in the plasma membrane [2,35]. Although the functions of SnRKs in plants have been deeply investigated, detailed functions could vary across different plant species. Therefore, it is necessary to dissect their functions in different types of plants.

*Dendrobium officinale* Kimura et Migo (also known as *D. catenatum* Lindl.) is an epiphytic plant in the family Orchidaceae. *D. officinale* has been used as a traditional medicinal plant in China and is also valued as an ornamental plant. *D. officinale* plants contain abundant polysaccharides as well as other metabolites, including alkaloids and bibenzyl compounds, which have been shown to be bioactive in immune modulation, as well as having anticancer and anti-oxidant properties, among others [36,37,38]. *D. officinale* plants prefer to grow in hot tropical areas and show strong resistance to stresses, particularly to drought [37]. Although SnRK members have been identified in various plants, such as *Arabidopsis* [10,39], apple [40], *Brachypodium distachyon* L. [41], *Eucalyptus grandis* [16], strawberry [12], and *Brassica napus* [42], there is no report on the SnRK family in orchid plants. In the present research, a comprehensive investigation of the SnRK family in *D. officinale* was carried out. A total of 36 DoSnRK members were identified, and their phylogenetic relationship was analyzed. The conserved domains and motifs, exon–intron structures, as well as the cis-elements in the promoter regions were characterized. The expression profiles of *DoSnRK* genes across eight tissues were analyzed based on transcriptomic data. The transcript levels of some *DoSnRK* genes under drought and hormone treatments were examined using quantitative real-time polymerase chain reaction (qRT-PCR). Therefore, the results provide important information on the SnRK family genes and pave the way for further understanding their functions in *D. officinale*.

## 2. Results

### 2.1. Genome-Wide Identification of DoSnRK Members in Dendrobium officinale

A total of 36 SnRK members, including 1 SnRK1, 7 SnRK2s, and 28 SnRK3s, were identified from the genome database of *D. officinale* using hidden Markov model (HMM). According to the conserved domains and nomenclature in *Arabidopsis*, they were named DoSnRK1.1, DoSnRK2.1 to DoSnRK2.7, and DoSnRK3.1 to DoSnRK3.28. Because the genome of *D. officinale* is presently assembled to the scaffold level, these 36 *DoSnRK* genes were distributed across 30 scaffolds. Basic information is presented in Table 1. The coding sequence (CDS) lengths of DoSnRKs are 1005–1530 bp. DoSnRK1.1 encodes the longest protein of 509 aa, while deduced proteins of DoSnRK2s have 334–387aa, and those of DoSnRK3 have 417–502 aa. The molecular weights of DoSnRK proteins are 37.87–58.01 kD, and isoelectric points are 4.77–9.37. The predicted subcellular localization showed that DoSnRK1.1 and most DoSnRK2 members were localized in the cytoplasm and nucleus, while most DoSnRK3 members were cytoplasmic, nuclear, and mitochondrial. Two members (DoSnRK3.10 and DoSnRK3.11) were at the endoplasmic reticulum.

### 2.2. Phylogenetic Relationship of DoSnRK Proteins

To analyze the phylogenetic relationship of SnRK proteins from *D. officinale* and *Arabidopsis*, an unrooted tree was constructed using MEGA 7.0 with the neighbor-joining (NJ) method (Figure 1). These 75 SnRK proteins were clearly grouped into three subfamilies: SnRK1, SnRK2, and SnRK3. The SnRK1 subfamily contained the fewest members, including one DoSnRK1 and three AtSnRK1 members. The SnRK2 subfamily consisted of 7 DoSnRK2 and 10 AtSnRK2 members, while The SnRK3 subfamily had the most members, including 28 DoSnRK3 proteins and 26 AtSnRK3 proteins. It was found that most of the DoSnRK proteins had corresponding homologs in *Arabidopsis*. The members of each subfamily from both species closely clustered, which suggests that these genes emerged before the two species diverged. Moreover, some SnRK members within species clustered together, implying expansion after speciation.

### 2.3. Gene Structures and Conserved Motifs of DoSnRK Genes

The gene structure could facilitate understanding of gene evolution. The exon-intron structures of 36 *DoSnRK* genes were analyzed with the GSGD. As shown in Figure 2, all the members of DoSnRK1 and DoSnRK2 subfamilies contained eight introns. Members from the DoSnRK3 subfamily varied in intron numbers and were divided into two types, one intron-free type and one intron-rich type. There were as many as 17 *DoSnRK3* genes without introns. Eight *DoSnRK3* genes contained 12–14 introns, and the rest of the three *DoSnRK3* genes harbored 1, 2, and 6 introns, respectively.

Multiple alignments of DoSnRK protein sequences showed that conserved domains were present (Figure 3). The highly conserved kinase domain was found in all the DoSnRK proteins, while less conserved and distinct domains were also found, including the ATP-binding domain and Domain I in DoSnRK2 members, while the NAF and PPI domains were found in the C-terminus of DoSnRK3 proteins.

To further reveal the structure of DoSnRK and AtSnRK proteins, 10 conserved motifs were searched with Multiple EM for Motif Elicitation (MEME). It was found that these 10 motifs were widely distributed in these proteins (Figure 4). Among them, eight motifs (motifs 1, 2, 3, 4, 5, 6, 9, and 10) were present in all the DoSnRK members, with an exception of DoSnRK3.6, which lacked motif 4. Among them, motifs 1, 2, 3, 4, 5, and 9 consisted of the protein kinase domain. DoSnRK1.1 contained nine motifs, without motif 8. All members of the DoSnRK2 subfamily had eight motifs and lacked motif 7 and motif 8. In the DoSnRK3 subfamily, 21 members contained 10 motifs, while 4 members did not include motif 8. Most members of AtSnRK proteins in the same subfamily contained the same motifs as DoSnRK proteins.

### 2.4. Cis-Elements in the Promoter Regions of DoSnRK Genes

The cis-element plays an important role in the control of gene expression. To explore the potential regulatory factors, cis-elements in 2 kb of the promoter regions of *DoSnRK* genes were predicted (Appendix A). The most frequent cis-elements were ethylene-responsive elements (ERE) and G-box, and they were present in the promoter regions of 27 and 28 *DoSnRK* genes, respectively, indicating that they could respond to ethylene and light. Other hormone-responsive elements such as CGTCA-motif, ABA-responsive element (ABRE), and TCA-element were found in the promoter regions of 25, 28, and 15 *DoSnRK* genes, respectively, which can respond to methyl jasmonate (MeJA), ABA, and salicylic acid, responsively. The other cis-elements involved in stress responses were also predicted in the promoter regions of *DoSnRK* genes, including anaerobic responsiveness element (ARE) in 28 *DoSnRKs*, the wound (WUN)-motif in 19 *DoSnRKs*, MYB binding site (MBS) in 15 *DoSnRKs*, and low temperature responsiveness (LTR) in 15 *DoSnRKs*. These cis-elements are responsible for stresses, including anaerobic, wounding, drought, and low temperature, respectively.

### 2.5. Expression Analysis of DoSnRK Genes in Eight Tissues

The expression profiles of 36 *DoSnRK* genes were investigated in eight tissues according to previous transcriptomic data. The tested tissues included the column, flower bud, lip, sepal, leaf, stem, white root, and green root tip. In orchid flowers, the column is the reproductive organ including the gynoecium (female organ) and androecium (male organ) and the lip is a highly specialized petal. Fragments Per Kilobase of transcript per Million Fragments (FPKM) were used to determine levels of gene expression. Three *DoSnRK* members (*DoSnRK3.6*, *DoSnRK3.20*, and *DoSnRK3.27*) were hardly detectable in any of the eight tissues due to very low FPKM values (less than 2), while other 33 *DoSnRK* genes were expressed in at least one tissue with FPKM values of more than 2. The heatmaps of *DoSnRK* genes were constructed using the R language (Figure 5). According to the expression patterns, these genes could be clustered into six groups.

Group I contained five members from the SnRK3 subfamily, including *DoSnRK3.12*, *DoSnRK3.16*, *DoSnRK3.18*, *DoSnRK3.24,* and *DoSnRK3.28*. They were mainly expressed in roots and flower tissues with low transcripts, while no transcript could be found in the stem or leaves. Most of them showed slightly higher expression in the white root than in the green root tip. *DoSnRK3.16* had abundant transcripts in the column and sepal, especially in the column with an FPKM of 239.56. Group II contained five members from the SnRK3 subfamily (*DoSnRK3.3*, *DoSnRK3.5*, *DoSnRK3.10*, *DoSnRK3.13,* and *DoSnRK3.14*). They showed moderate expression levels in various tissues with FPKM values of less than 25.89. Group III consisted of six members, including *DoSnRK2.3* and five members from the SnRK3 subfamily. These six genes were expressed in all the eight tissues, and most of the members showed medium transcripts and the FPKM was between 5.67 and 54.06. Group IV contained 15 members, including *DoSnRK1.1*, 6 from the SnRK2 subfamily, and 8 from the SnRK3 subfamily. Most genes from this group showed extensive expression in eight tissues, while a few genes were lowly expressed in certain tissues. Group V included three genes (*DoSnRK3.6*, *DoSnRK3.*7, and *DoSnRK3.17*). *DoSnRK3.17* was expressed only in the lip tissue with a low level and not expressed in other tissues. The remaining two genes were not expressed or expressed with very low transcripts in all the tissue. Group VI had two members, *DoSnRK3.20* and *DoSnRK3.27*, that were not expressed in any tissue.

To further confirm the expression patterns of *DoSnRK* genes from the RNA sequencing, 10 genes and 3 tissues were selected to perform qRT-PCR assays (Appendix A). The results showed that the expression patterns from two methods exhibited similar results. All these genes could be detected in roots, stems, and leaves. For example, *DoSnRK2.3* was expressed the highest in roots and the lowest in leaves. However, there was a slight difference in *DoSnRK2.2* between the two analysis methods.

### 2.6. DoSnRK Gene Expression after Drought, MeJA, and ABA Treatments

To investigate the potential function of DoSnRKs in response to exogenous factors, three treatments were carried out on *D. officinale* plants, including drought, MeJA, and ABA. The expression levels of 27 *DoSnRK* genes were detected using qRT-PCR assays. The results are shown in Figure 6.

After drought treatment, 2 genes (*DoSnRK2.7* and *DoSnRK3.23*) showed no change in transcripts, while the other 25 *DoSnRK* genes showed various changes at the transcriptional level. Among them, six *DoSnRK* genes showed decreased expression levels, while eight *DoSnRK* genes showed increased expression levels. Eight *DoSnRK* genes exhibited decreased or increased changes 3 h after treatment but recovered to the control level 6 h after treatment. Two *DoSnRK* genes (*DoSnRK3.21* and *DoSnRK3.24*) did not show transcriptional changes until 6 h after treatment. In particular, the expression of *DoSnRK3.8* gene increased at 3 h and decreased at 6 h but continued to remain higher than the control.

After MeJA treatment, 3 genes (*DoSnRK3.2*, *DoSnRK3.5,* and *DoSnRK3.23*) had no response at the transcriptional level, while the other 24 *DoSnRK* genes exhibited differential expression changes. There were 3 and 11 *DoSnRK* genes that showed increased and decreased transcripts, respectively. Four *DoSnRK* genes were responsive to treatment at 3 h but eventually returned to levels similar to the control. Three genes did not respond to the treatment until 6 h after treatment. The expression levels of *DoSnRK2.6* and *DoSnRK3.8* increased at 3 h but decreased at 6 h.

After ABA treatment, four *DoSnRK* genes kept similar transcripts as those in the control, four *DoSnRK* genes showed increased expression, while seven genes showed decreased expression. Moreover, the transcripts of five *DoSnRK* genes changed at 3 h but eventually returned to the control level. Four genes were responsive to ABA 6 h after treatment. Notably, the expression of the *DoSnRK2.6* gene sharply increased at 3 h but was lower than the control at 6 h.

It was found that some genes could respond to multiple treatments. For example, the expression levels of *DoSnRK1.1* were down-regulated by three treatments, and the transcripts of *DoSnRK2.2* were enhanced by three treatments. These results showed their specific expression changes. To dissect the relationship between the transcriptional change and the cis-elements in the promoter region, the cis-elements in the stress-responsive *DoSnRK* genes were investigated. It was found that 10 members of 25 drought-responsive *DoSnRK* genes contained the MBS-binding site that is responsive to drought, while 16 members of 24 drought-responsive *DoSnRK* genes contained the CGTCA-motif that is responsible for the response to MeJA, and 15 members of 23 ABA-responsive *DoSnRK* genes included the ABRE motif that can respond to ABA. These results suggest that these cis-elements in their promoter regions of *DoSnRK* genes play a role in response to stress.

## 3. Discussion

As one type of Ser/Thr protein kinases, the SnRK family has been demonstrated to function in plant metabolism, signaling transduction, and stress response [2,7]. In the present research, SnRK members were first comprehensively identified from the genome of *D. officinale*, which will provide useful information to investigate their functions in the future.

A total of 36 DoSnRK members were identified in *D. officinale*. Compared with the number of SnRK members in other species, the number in *D. officinale* was similar to that in *Arabidopsis* and in *Eucalyptus grandis* [16], but *D. officinale* has 10 more than strawberry [12] and fewer than *Brassica napus* [42] and *Brachypodium distachyon* [41]. Among the three subfamilies, the DoSnRK1 subfamily had only one member (DoSnRK1.1), which is the same as that in strawberry [12] but less than that in other species. The DoSnRK2 subfamily contained moderate members, as in sugar beet [13], tea [44], *E. grandis* [16], and strawberry [12], but some species contained more than 10 members [41,42,45,46,47,48,49]. The DoSnRK3 subfamily contained the most members, similar to *B. distachyon* [41] and *E. grandis* [16], but fewer than *Brassica. napus* [42]. These results support the previous view that the SnRK2 and SnRK3 subfamilies evolved from the SnRK1 subfamily and increased their number during evolution [12].

The analyses of gene structures and conserved motifs showed that DoSnRKs exhibited high conservation during evolution. *DoSnRK1.1* contained eight introns, fewer found than in strawberry [12], *E. grandis* [16], and *B. distachyon* [41]. The entire DoSnRK2 subfamily included eight introns, as most SnRK2 members from other species such as *Arabidopsis* [10] and rice [11]. The DoSnRK3 subfamily included the intron-free type and the intron-rich type, which was found in strawberry [12]. These results suggest that the SnRK2 subfamily contained the same number of introns as in the SnRK1 family, while some members from the SnRK3 subfamily lost introns during evolution.

The functions of genes could be implied based on their expression. In the study, as many as 26 members of 36 *DoSnRK* genes were expressed in multiple tissues. The ubiquitous expression pattern of SnRK genes has been found in other species [41]. For example, there were eight *SnRK2* genes that were widely transcribed in a variety of tissues in *Brachypodium distachyon* [41]. The extensive expression of *SnRK* genes indicates that SnRKs could be necessary for plants to grow and develop different tissues (root, stem, leaf, flower, etc.). For the same subfamily gene from different species, there is a difference in their expression. For example, *FvSnRK1.1* in strawberry was specifically expressed in certain tissues [12], while *DoSnRK1.1* was abundantly expressed in different tissues, which could be species specific.

Previous studies have found that *SnRK* genes can respond to a variety of environmental stresses at the transcriptional level [40,50]. In *B. distachyon*, there were nine *SnRK2* genes that were induced by ABA [41]. Similarly, multiple *SnRK2* genes in *Arabidopsis* were responsive to ABA [25]. In our studies, it was found there were a total of 23 *DoSnRK* genes that could respond to ABA. Moreover, some *SnRK* genes were found to be responsive to multiple stresses as in *D. officinale*. These different SnRK subfamilies and different members from them showed distinct changes after different treatments. Based on expression changes under three treatments, it was found that 27 *DoSnRK* genes could be classified into three types. The first type had the lowest fold changes, including 19 members from all the three DoSnRK subfamilies. Most of them showed less than onefold increase or decrease after at least one treatment. The second type contained six members from the DoSnRK2 and DoSnRK3 subfamilies, and they showed medium fold changes (more than onefold but less than fivefold changes). The third type included two members (*DoSnRK2.6* and *DoSnRK3.16*) that showed sharp fold changes after treatment. For them, as many as more than 150-fold changes were detected after MeJA and ABA treatment. In *B. napus*, extreme expression changes of *BnSnRK2.24* after treatments including ABA were also observed [42]. Other *SnRK* genes from other species responding to various stresses were also reported, suggesting their similar functions in response to environmental changes. However, certain *SnRK* genes exhibiting distinct responses to different treatments have indicated the specificity of SnRK during evolution [41,42,48].

## 4. Materials and Methods

### 4.1. Plant Materials

Healthy and well-grown in vitro plants of *D. officinale* were kept by tissue culture in our lab. The basic medium was Murashige and Skoog (MS) medium [51] (Solarbio, Beijing, China) supplemented with 30 g/L sucrose, and plants were grown with 12 h light/12 h dark at 25 °C. For RNA extraction, the leaves, stems, and roots from three six-month-old plants were sampled, immediately frozen in liquid nitrogen, and kept at −80 °C until use.

### 4.2. Plant Treatments

Two kinds of hormone treatments were performed on six-month-old plants. MeJA (Coolaber, Beijing, China) and ABA (Sangon, Shanghai, China) were dissolved in ethanol, and then deionized water was added to produce a 10 mM stock solution. The stock solution was diluted to a concentration of 100 µM. For each treatment, at least three plants were sprayed with MeJA or ABA for three or six hours, while the control plants were sprayed with the corresponding solution without MeJA or the ABA reagent. For drought treatment, plants were taken out of the medium and put on filter paper for three or six hours. The leaves from treatment and control plants were harvested, immediately frozen, and saved at −80 °C for RNA purification.

### 4.3. Identification of the SnRK Family in D. officinale

The protein sequences of *D. officinale* were obtained from the National Center for Biotechnology Information (NCBI) [52]. The protein kinase domain (PF00069) was used to search potential DoSnRK proteins using HMMER [53]. The protein kinase domain was further confirmed using SMART (http://smart.embl-heidelberg.de/, accessed on 3 October 2020) and CDD (https://www.ncbi.nlm.nih.gov/cdd, accessed on 3 October 2020). Redundant and incomplete sequences were removed, and the remaining sequences containing PF00069 were considered as candidate SnRK members. The molecular weight point and isoelectric point (pI) of deduced DoSnRK proteins were calculated using ProtParam [54]. The possible subcellular localization of DoSnRKs was forecasted using WOLF PSORT (https://www.genscript.com/psort.html, accessed on 3 October 2020).

### 4.4. Phylogenetic Relationship of DoSnRK Proteins

The protein sequences of AtSnRKs in *Arabidopsis* were acquired from The Arabidopsis Information Resource (TAIR). To analyze the phylogenetic relationship between DoSnRK proteins and AtSnRK proteins, their full-length sequences were aligned using ClustalW [55], and the phylogenetic tree was constructed using MEGA 7.0 with a neighbor-joining (NJ) method and a bootstrap of 1000 replicates [56].

### 4.5. Exon–Intron Structures and Conserved Motifs in DoSnRK Genes

The exon–intron structures of 36 *DoSnRK* genes were analyzed using the Gene Structure Display Server (GSDS) program by comparing the coding sequences (CDS) and the genomic sequences [57]. The conserved motifs in DoSnRK and AtSnRK proteins were searched using Multiple Em for Motif Elicitation (MEME) [58], and motifs were further analyzed using the InterPro database [59].

### 4.6. Expression Profiles of DoSnRK Genes in Eight Tissues

The expression levels of *DoSnRK* genes in eight *D. officinale* tissues were estimated according to the published RNA sequencing data [43]. The FPKM values were calculated as in our previous method in these tissues, including the column, flower bud, lip, sepal, leaf, stem, white root, and green root tip [60,61]. An expression heatmap of *DoSnRK* genes was produced using the R package pheatmap, v1.0.10 (https://rdrr.io/cran/pheatmap/, accessed on 13 January 2021).

### 4.7. qRT-PCR Assays

Total RNA from each sample was purified using the RNAprep Pure Plant Plus Kit (Cat. DP441, Tiangen, Beijing, China) as per the instructions. For cDNA synthesis, the concentration and quality of total RNA samples were tested on the machine Nanodrop 2000 (Thermo Scientific, Waltham, MA, USA). Next, 200 ng of total RNA for each sample was used to prepare cDNA in the same reaction system of 20 µL using the cDNA Synthesis kit (Cat. AT341, Transgen, Beijing, China). The SuperReal PreMix Plus SYBR Green Kit (Tiangen, Beijing,China) was used to perform qRT-PCR. In brief, 1 µL of the cDNA mixture was mixed with SYBR Green mix in 20 µL of the reaction system. qRT-PCR was performed on the Applied Biosystems QuantStudio 6 Flex Real-Time PCR System (ThemoFisher Scientific, Waltham, MA, USA). The primers for qRT-PCR assays of *DoSnRK* genes were designed on Primer3 (https://bioinfo.ut.ee/primer3-0.4.0/, accessed on 13 October 2020) and further aligned with the *DoSnRK* sequences to make specific sequences at the 3’-end. The sequences of primers used in qRT-PCR are listed in Appendix A. The PCR conditions were as follows: pre-denaturation at 95 °C for 15 min and then 40 cycles at 95 °C for 10 s, at 60 °C for 20 s, and finally at 72°C for 30 s. The melt curves were carried out at 95 °C for 15 s, at 60 °C for 1 min, and at 95 °C for 15 s. The *DoActin* gene was used as an internal reference. The levels of gene expression were calculated using the 2^−ΔΔ*CT*^ method [62], and the values of CT represent an average of cycle times from three repeats.

### 4.8. Cis-Element Analysis in Promoter Regions of DoSnRK Genes

To analyze the possible cis-elements in the promoter regions of *DoSnRK* genes, sequences of 2 kb upstream of the translation start site were obtained from the NCBI and submitted to PlantCARE to search.

## 5. Conclusions

This study presents the first instance of comprehensive information about SnRK family genes in *D. officinale*. The phylogenetic analysis showed that 36 DoSnRKs could be grouped into three subfamilies (DoSnRK1, DoSnRK2, and DoSnRK3) with conserved domains and motifs. The expression analyses of *DoSnRK* genes in eight tissues showed that multiple *DoSnRK* genes were ubiquitously expressed. Drought, MeJA, and ABA treatments could induce transcriptional changes in most *DoSnRK* genes. Therefore, our results provide valuable information about *DoSnRK* genes for further understanding the functions of SnRKs in plants.

## Figures and Tables

**Figure 1 plants-10-00479-f001:**
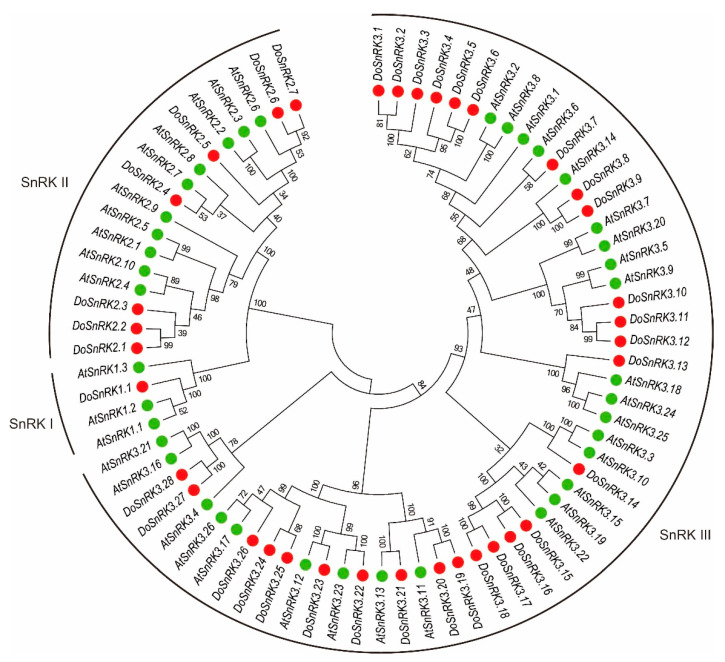
Phylogenetic analysis of sucrose non-fermenting1-related protein kinase (SnRK) proteins from *D. officinale* and *Arabidopsis*.

**Figure 2 plants-10-00479-f002:**
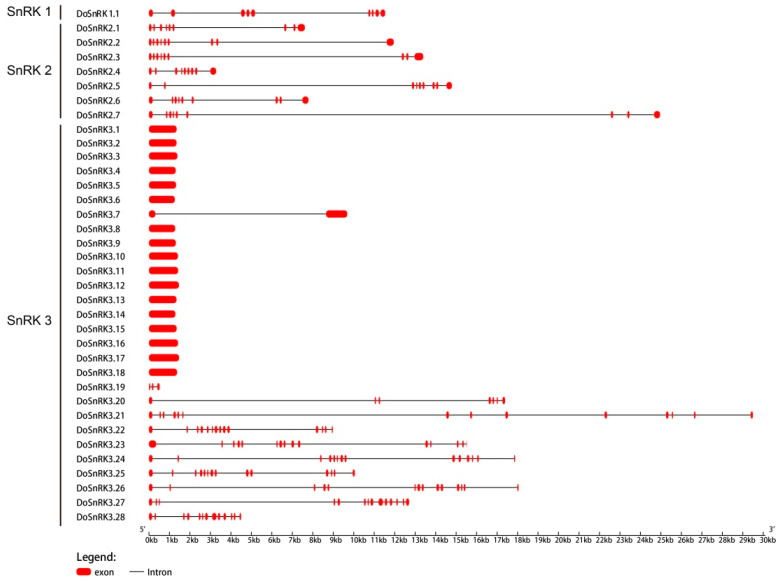
Gene structures of *DoSnRK* genes in *D. officinale*.

**Figure 3 plants-10-00479-f003:**
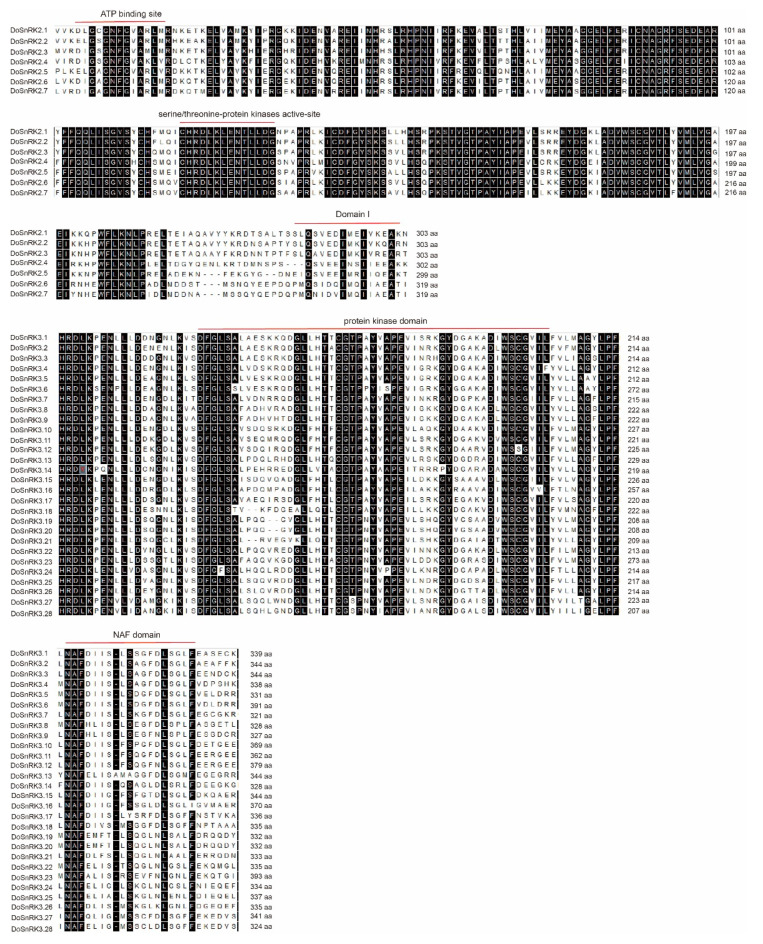
The multiple sequence alignment of DoSnRK2 and DoSnRK3 proteins.

**Figure 4 plants-10-00479-f004:**
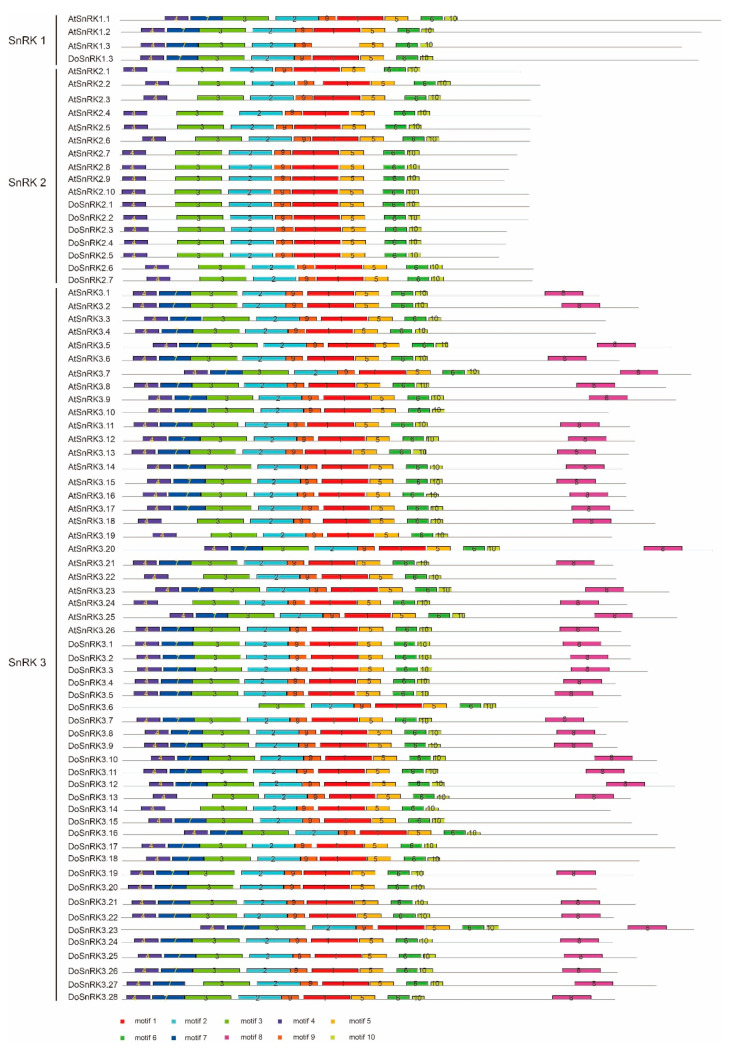
The conserved motifs of SnRK proteins predicted using Multiple Em for Motif Elicitation (MEME). The colored rectangles with numbers stand for 10 conserved motifs/ and the gray lines are non-conserved sequences.

**Figure 5 plants-10-00479-f005:**
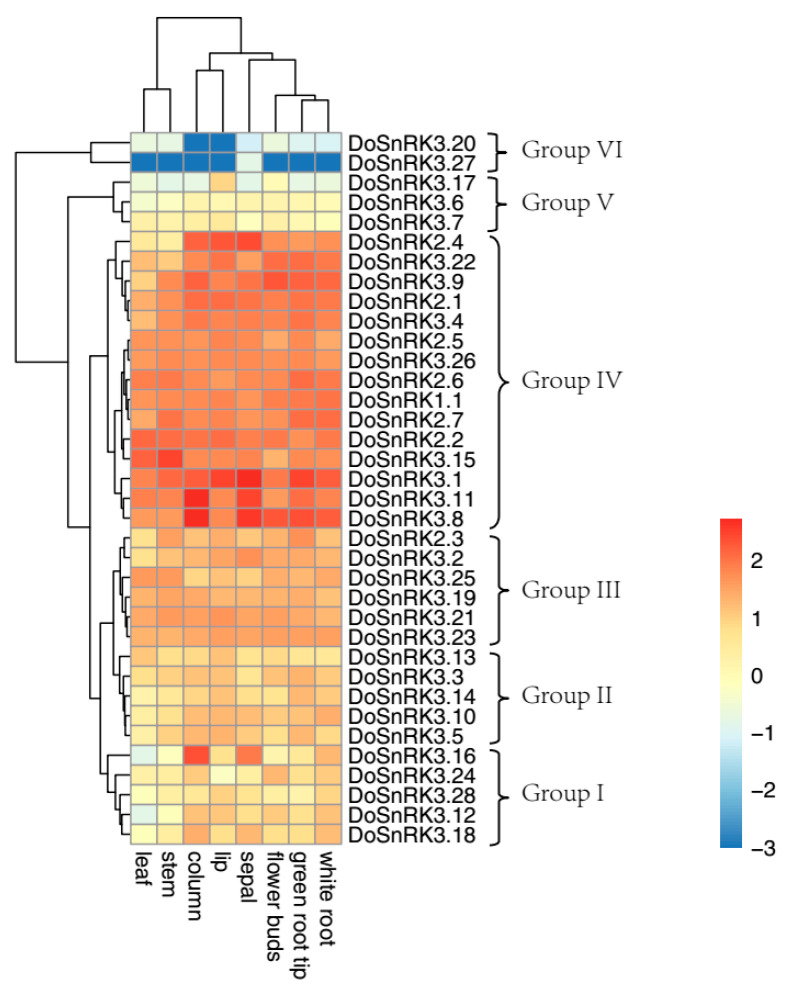
Expression patterns of 36 *DoSnRK* genes in eight tissues of *D. officinale*. The expression levels were based on the RNA sequencing data [43]. The eight tissues tested included the column, flower bud, lip, sepal, leaf, stem, white root, and green root tip. The color scale stands for the values of log_2_(FPKM + 0.001).

**Figure 6 plants-10-00479-f006:**
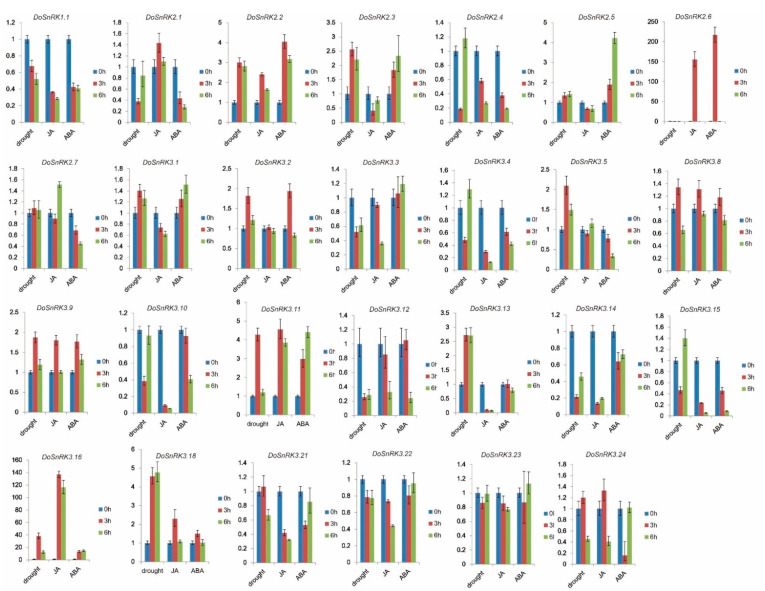
Expression levels of 27 *DoSnRK* genes under three treatments using qRT-PCR assays. The mock-treated leaves at the beginning (0 h) were used as the control to calculate the relative expression levels at 3 and 6 h after treatment. Each experiment was performed with at least three individual plants with three replications. The bars denote the standard deviation.

**Table 1 plants-10-00479-t001:** The list of 36 *DoSnRK* genes identified in *Dendrobium officinale*.

Gene Name	Gene ID	Scaffold	Length of Coding Sequences (CDS) (bp)	Protein Length (aa)		Mol. Wt. (kDa)	pI	Location
DoSnRK1.1	LOC110093905	737	1530	509		58.01	8.77	a,b
DoSnRK2.1	LOC110104807	231	1086	361		41.6	5.81	a,b
DoSnRK2.2	LOC110107470	1105	1080	359		41.39	5.58	a,b
DoSnRK2.3	LOC110094403	209636	1164	387		44.66	5.17	b,a
DoSnRK2.4	LOC110102148	211	1026	341		39.04	5.62	a,b
DoSnRK2.5	LOC110092671	526	1005	334		37.87	5.69	a,b
DoSnRK2.6	LOC110112577	459	1092	363		41.17	4.90	a,c
DoSnRK2.7	LOC110116324	601	1086	361		40.92	4.77	c,a
DoSnRK3.1	LOC110116107	456	1350	449		50.97	9.25	b,c,a
DoSnRK3.2	LOC110099074	313	1347	448		50.97	9.07	b,c,a
DoSnRK3.3	LOC110112518	1217	1389	462		52.43	9.08	c,a
DoSnRK3.4	LOC110112938	61322	1305	434		49.76	9.18	a,c,b
DoSnRK3.5	LOC110100634	1056	1323	440		50.23	9.16	b,a,c
DoSnRK3.6	LOC110104808	231	1263	420		46.87	9.37	c,a
DoSnRK3.7	LOC110113686	624	1341	446		49.92	9.37	b,c,a
DoSnRK3.8	LOC110094439	629	1281	426		47.15	9.04	a,b
DoSnRK3.9	LOC110093407	1038	1311	436		48.19	9.10	d,c,b
DoSnRK3.10	LOC110095265	768	1416	471		53.46	8.79	d
DoSnRK3.11	LOC110113865	370	1422	473		53.5	7.99	d
DoSnRK3.12	LOC110109095	117410	1461	486		54.74	8.01	b,a
DoSnRK3.13	LOC110093408	1038	1344	447		49.78	8.59	d,c,b
DoSnRK3.14	LOC110095499	827	1290	429		48.2	9.32	b,a
DoSnRK3.15	LOC110102687	123	1350	449		50.5	8.28	b,a,c
DoSnRK3.16	LOC110098084	313	1419	472		52.82	9.30	a,b,c
DoSnRK3.17	LOC110100633	1056	1467	488		54.69	9.10	a,b,c
DoSnRK3.18	LOC110112951	61322	1371	456		50.94	8.80	a,b,c
DoSnRK3.19	LOC110105714	260	1353	450		51.06	8.37	a,b
DoSnRK3.20	LOC110098208	231	1254	417		47.59	5.91	a,c
DoSnRK3.21	LOC110113559	1557	1350	449		51.09	8.01	a
DoSnRK3.22	LOC110110059	1220	1299	432		48.82	8.94	b,c,a
DoSnRK3.23	LOC110101878	974	1509	502		57.14	9.24	b,c,a
DoSnRK3.24	LOC110103756	716	1293	430		48.82	8.87	a,b,c
DoSnRK3.25	LOC110113526	1721	1356	451		51.47	8.70	b,a,c
DoSnRK3.26	LOC110114482	1608	1305	434		49.36	8.32	b,c,a
DoSnRK3.27	LOC110112680	627	1407	468		52.97	7.51	b,c,a
DoSnRK3.28	LOC110107680	142	1299	432		48.78	8.01	b,a,c

a, Cytoplasmic; b, nuclear; c, mitochondrial; d, endoplasmic reticulum.

## Data Availability

Publicly available datasets were analyzed in this study. This data can be found in NCBI (PRJNA348403).

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
