# Peer review of "Genomic Characterization and Expression Analysis of the SnRK Family Genes in *Dendrobium officinale* Kimura et Migo (Orchidaceae)"

_plants, 2021, doi:10.3390/plants10030479_

Round 1
Reviewer 1 Report
The manuscript “Genomic Characterization and Expression Analysis of the SnRK family genes in Dendrobium officinale” presents a comprehensive analysis of the genes of the family of sucrose nonfermenting-1-related protein kinases (SnRKs) in Dendrobium officinale related to the protein primary and secondary structure, promoter cis acting element analysis, and gene expression analysis in different tissues and under conditions of drought and hormonal treatments with MeJA and ABA.
The experiments seem properly executed and the analysis of data is sound. The results are of interest and the information regarding genes encoding SnRKs and there tissue specificity and response to external factors will be useful for the understanding of their functional roles and the aspects of family member divergence and member functional specificity.
However certain issues need to be addressed before manuscript acceptance:
- English has to be improved throughout the manuscript.
- The discussion of the expression analysis by qPCR could be much more elaborate. Authors have gathered a wealth of information regarding transcript abundance of a multitude of gene members and under three different conditions, drought, JA and ABA treatments.
-The -fold change in transcript accumulation between treatments and among different genes should be discussed. How do different families and different members compare to each other?
For example, remarkable changes are observed in the DoSnRK2.6 where there is approximately 150x and 200x increase in transcript accumulation upon JA and ABA treatment, respectively. What would this suggest and how does it relate to the literature? Equally, for the DoSnRK3.16.
-In addition, given the high number of gene family members and their relatedness, authors should provide better detail of primer design used for qRT-PCR analysis. Which regions (non-conserved) were used to design gene specific primers in order to distinguish among the different genes and provide credible gene expression results?
Other:
Line 53: explain what are NAF and PPI (full names)
Line 69: replace ‘which’ with ‘and it was’
Line 101-102: references??
Line 113: ‘pave the foundation for the further..’ please replace with ‘pave the way for further..)
Line 189: what are column and lips?
Line 275: specify ‘was more’
Line 277: specify which member?
Line 300: what is Fv?
Line 328: provide links
Line 335: ‘to analysis’ change to ‘to analyze’
Reviewer 2 Report
Comments have been directly added in the attached pdf
